# Gradient-Based Program Synthesis with Neurally Interpreted Languages

**Matthew V. Macfarlane**[*]
AMLab, University of Amsterdam
matthew.v.m@live.co.uk

**Clément Bonnet**
Ndea

**Herke van Hoof**
AMLab, University of Amsterdam

**Levi H. S. Lelis**
Department of Computing Science, University of Alberta
Alberta Machine Intelligence Institute (Amii), Edmonton, Canada

## Abstract

A central challenge in program induction has long been the trade-off between symbolic and neural approaches. Symbolic methods offer compositional generalisation and data efficiency, yet their scalability is constrained by formalisms such as domain-specific languages (DSLs), which are labour-intensive to create and may not transfer to new domains. In contrast, neural networks flexibly learn from data but tend to generalise poorly in compositional and out-of-distribution settings. We bridge this divide with an instance of a Latent Adaptation Network architecture named Neural Language Interpreter (NLI), which learns its own discrete, symbolic-like programming language end-to-end. NLI autonomously discovers a vocabulary of primitive operations and uses a novel differentiable neural executor to interpret variable-length sequences of these primitives. This allows NLI to represent programs that are not bound to a constant number of computation steps, enabling it to solve more complex problems than those seen during training. To make these discrete, compositional program structures amenable to gradient-based optimisation, we employ the Gumbel-Softmax relaxation, enabling the entire model to be trained end-to-end. Crucially, this same differentiability enables powerful test-time adaptation. At inference, NLI's *program inductor* provides an initial program guess. This guess is then refined via gradient descent through the *neural executor*, enabling efficient search for the neural program that best explains the given data. We demonstrate that NLI outperforms in-context learning, test-time training, and continuous latent program networks on tasks that require combinatorial generalisation and rapid adaptation to unseen tasks. Our results establish a new path toward models that combine the compositionality of discrete languages with the gradient-based search and end-to-end learning of neural networks.

## 1 Introduction

A central challenge in machine learning is the trade-off between symbolic and neural representations. Symbolic approaches often enable strong compositional generalisation (Lake & Baroni, 2018), in some cases from only a few examples (Solar-Lezama et al., 2006; Gulwani, 2011). Yet their scalability is constrained by formalisms such as domain-specific languages, which require human effort to generate, may not transfer to other domains, and are combinatorially expensive to search. Neural approaches, by contrast, scale effectively but behave as monolithic models. The knowledge they acquire is entangled in their weights, making it difficult to reuse beyond the training distribution, even when generalisation only requires recombining concepts already learned (Baroni, 2020).

In the context of program synthesis, we make progress toward bridging this divide with a model that learns its own symbolic representation, end-to-end, directly from data. Specifically, it simultaneously learns a domain-specific neural language and a neural interpreter for such a language. Recent work

---

[*]Work completed while a visiting student at the University of Alberta.

has shown the power of Latent Adaptation Networks (LANs) through the Latent Program Network architecture (LPN) (Macfarlane & Bonnet, 2025). We refer to such models as LANs, as they are encoder-decoder architectures that learn a latent space to represent the model space. The key feature of these architectures is that at test time, the latent space is searched to perform adaptation.

LPNs have shown promise with simple continuous latent spaces and single forward pass conditioning on the continuous latent embedding for prediction. They have promising properties like resistance to overfitting due to the compressed space and the ability to perform rapid adaptation in domains such as program repair (Silva et al., 2025). However they have been shown to have limited abilities for compositional generalisation (Macfarlane & Bonnet, 2025).

We introduce the Neural Language Interpreter architecture (NLI), that also is a Latent Adaptation Network but instead leverages discrete representations (Jang et al., 2017; Maddison et al., 2017) to represent programs. To learn a discrete vocabulary that can represent programs in the target domain, we train on programming-by-example (PBE) tasks. During inference, conditioned on a specification of examples, NLI's encoder acts as a program inductor, producing a sequence of discrete tokens that forms a neural program. The decoder serves as a neural executor, interpreting the program one token at a time, mapping the test input to an output, similar to neural executors used in conditional world models (Ha & Schmidhuber, 2018). Both the encoder and decoder are designed to be fully differentiable, so NLI can be trained end-to-end.

Since the neural executor consumes one token at a time, NLI is not bound to a fixed number of computational steps, as in LPN. The number of steps in NLI's programs can grow with the token length of programs. This is important because it allows NLI to solve problems more challenging than those with constant-time requirements, seen in training. Moreover, since NLI's programs can recombine learned tokens in different ways and at different lengths, we hypothesise that its language supports combinatorial generalisation.

In addition to the engineering hurdle of designing domain-specific languages, our work is motivated by the need to bypass the difficult combinatorial search problem inherent to program synthesis. Rather than learning external guiding functions for search (Barke et al., 2020; Odena et al., 2021; Ameen & Lelis, 2023), NLI embeds guidance in the language it learns. Since the neural executor is differentiable, we can search in the space of neural programs with gradient descent. Synthesising a neural program with NLI is thus analogous to local search in symbolic spaces (Husien & Schewe, 2016), but with the advantage of having gradient signals. Another benefit of a learned language is how the search is initialised, which can dramatically affect efficiency (Hoos & Stützle, 2004; Sadmine et al., 2024). NLI's inductor provides an initial guess for a neural program solution at test time, and the gradient search refines this guess to find a neural program that solves the problem. Across sequence-based compositional benchmarks, NLI achieves strong out-of-distribution accuracy on length extrapolation, primitive extraction, and novel composition tasks, where in-context learning, test-time training, and latent program networks fail. NLI demonstrates competitive performance compared to neuro-symbolic baselines on DeepCoder with access to ground truth program representations, despite training only from input–output examples.

## 2 PROBLEM STATEMENT

We formalise our task as **program induction**, where the goal is to infer the underlying behaviour of an unknown program $p$ from input-output examples using a model $M$. Given a set $S = \{(x_i, y_i)\}_{i=1}^n$ of $n$ input-output pairs generated by $p$ and a new query input $x_{n+1}$, $M(S, x_{n+1})$ predicts the corresponding output $p(x_{n+1})$. This aligns with the programming by example (PBE) formalisation, where information about program $p$ is available only via its outputs. Training tasks are formed by sampling a latent program $p$ from a distribution $P_{\text{train}}$ over the space of possible programs $\mathcal{P}$. Program specifications are formed from $n+1$ inputs sampled from the program-dependent conditional distribution $\{x_i\}_{i=1}^{n+1} \sim P(X|p)$. This distribution generates inputs relevant to the logic of program $p$. The first $n$ input-output pairs form the specification $S = \{(x_i, p(x_i))\}_{i=1}^n$, from which the model must induce the program's logic. The model's objective is to minimise the prediction error between its prediction, $\hat{y}_{n+1} = M(S, x_{n+1})$, and the true output $p(x_{n+1})$. This is to train the model to generalise program execution to a new input, and not merely fit the training pairs. The model has no access to the program's fully observable representation $p$ during training or test time. This is vital, as

real-world tasks often involve latent $p$ without an observable specification. At test time, the model is evaluated on programs drawn from $P_{\text{test}}$, which can differ from $P_{\text{train}}$, to test for generalisation.

# 3 DISCRETE SEQUENTIAL INFERENCE

Existing neural program synthesis methods fail at compositional generalisation, struggling to recombine learned concepts for novel tasks (Macfarlane & Bonnet, 2025). The Neural Language Interpreter (NLI) addresses this by learning a discrete, symbolic-like programming language end-to-end, representing programs as variable-length token sequences executed by a differentiable neural executor. Program representations are learned purely from input-output examples using Gumbel-Softmax (Jang et al., 2017; Maddison et al., 2017), which also enables efficient gradient-based search at test time.

## 3.1 TRAINING OBJECTIVE

In order to learn a latent space which represents the space of possible programs we make use of an encoder-decoder architecture. The encoder (program inductor $q_\phi$) infers a latent program representation from a specification of input-output pairs, which the decoder (neural interpreter $p_\theta$) then executes to predict outputs for a given input. This architecture is trained with an objective that encourages the encoder to produce latent representations that are both faithful to the specification and generalisable to held-out pairs. Formally, NLI is trained on mini-batches of size $B$, where each specification contains $m$ input-output pairs. NLI uses a leave-one-out loss, where the program inductor $q_\phi$ induces a latent representation using $m-1$ pairs and the prediction error is computed for the input-output pair left out. Namely, for the $b$-th specification in the mini-batch and the $i$-th input-output pair of that specification, we define $\mathcal{S}_{b,i} = \mathcal{S}_b \setminus \{(x_{b,i}, y_{b,i})\}$. For each $\mathcal{S}_{b,i}$, we maximise the likelihood of predicting $y_{b,i}$ from $x_{b,i}$, while regularising for the reuse of learned neural tokens. The objective is defined as follows.

$$\mathcal{L}(\phi, \theta; \mathcal{S}) = \frac{1}{B} \sum_{b=1}^{B} \left( \frac{1}{m} \sum_{i=1}^{m} \mathcal{L}_{\text{recon}}(\phi, \theta; x_{b,i}, y_{b,i}, \mathcal{S}_{b,i}) + \lambda_{\text{reg}} \cdot \mathcal{L}_{\text{reg}}(\phi; \mathcal{S}_b) \right) . \tag{1}$$

This objective is formed of the reconstruction loss ($\mathcal{L}_{\text{recon}}$) and the encoder regularisation loss ($\mathcal{L}_{\text{reg}}$).

**Reconstruction Loss ($\mathcal{L}_{\text{recon}}$)** This term ensures that the latent program is expressive enough to predict the program's output on the held-out input. $\mathcal{L}_{\text{recon}}$ is defined as the negative log-likelihood of the target output $y$ given the input $x$ and the latent program $z$ inferred from the specification $S$.

$$\mathcal{L}_{\text{recon}}(\phi, \theta; x, y, S) = -\log p_\theta(y \mid x, q_\phi(S)) . \tag{2}$$

**Encoder Regularisation Loss ($\mathcal{L}_{\text{reg}}$)** This regularising loss encourages reuse of tokens in the neural vocabulary of size $K$, biasing the encoder (via parameters $\phi$) toward discovering a compositional latent program space. We implement a differentiable approximation of the number of unique tokens used in the batch. By penalising programs that use many unique vocabulary entries, $\mathcal{L}_{\text{reg}}$ promotes generalisation: the model learns to build new programs by recombining a compact set of discovered primitives rather than representing each new task with a single new token.

$$\mathcal{L}_{\text{reg}}(\phi; \mathcal{S}) = \sum_{k=1}^{K} \sum_{j=1}^{N} \left[ 1 - \exp\left( \sum_{b=1}^{B} \sum_{i=1}^{m} \log\left(1 - q_\phi^{(k,j)}(\mathcal{S}_{b,i})\right) \right) \right] . \tag{3}$$

Here, $N$ is the number of token positions in the program and $q_\phi^{(k,j)}(\mathcal{S}_{b,i})$ is the probability assigned to token $k$ at position $j$ by the encoder. The $\exp$ term corresponds to the probability that token $k$ is not used at any position $j$ across the leave-one-out specifications in the batch. Thus, the $1 - \exp$ term is the probability that token $k$ is used at least once in the batch. Summing this quantity over all tokens $k$ yields a differentiable approximation to the expected number of unique tokens in the batch.

## 3.2 INDUCTOR: DISCRETE PROGRAM REPRESENTATION LEARNING

The encoder $q_\phi$ defines a mapping from a specification $S \subseteq \mathcal{X} \times \mathcal{Y}$, a finite set of input-output examples, to a latent program representation $\mathbf{z} = (z_1, \ldots, z_T)$, where each $z_t \in \Delta^{K-1}$ is a distribution

over $K$ codebook entries; $q_\phi : \mathcal{S} \to (\Delta^{K-1})^T$. The codebook includes a dedicated skip token, which acts as an instruction to the decoder to perform a no-op, allowing the model to represent programs shorter than the fixed sequence length $T$ by ignoring unnecessary computational steps.

The encoder operates in two steps. First, a transformer $h_\phi : \mathcal{X} \times \mathcal{Y} \to \mathbb{R}^{T \times d}$ is applied independently to each pair $(x_i, y_i) \in S$, producing a sequence of $T$ token embeddings. The specification embedding is then obtained by mean-pooling each token position across all pairs in $S$:

$$h_\phi(x_i, y_i) = \left(e_1^{(i)}, \ldots, e_T^{(i)}\right), \qquad \bar{e}_t = \frac{1}{|S|} \sum_{i=1}^{|S|} e_t^{(i)}, \qquad \bar{e} = \left(\bar{e}_1, \ldots, \bar{e}_T\right) \in \mathbb{R}^{T \times d}$$

This aggregation scheme is permutation-invariant with respect to the ordering of pairs, a similar method to that used in Neural Processes (Garnelo et al., 2018). The aggregation also helps generalise to different specification sizes at test time from those seen in training (Macfarlane & Bonnet, 2025).

In the second step, each embedding $\bar{e}_t$ is projected to a distribution over the codebook. A shared feed-forward network $f_\phi$ maps $\bar{e}_t$ to a vector of logits over $K$ entries, and a Gumbel-Softmax relaxation (Jang et al., 2017; Maddison et al., 2017) yields a differentiable approximation of a discrete sample:

$$l_t = f_\phi(\bar{e}_t) \in \mathbb{R}^K, \qquad z_t = \mathrm{Softmax}\left(\frac{l_t + g_t}{\tau_e}\right) \in \Delta^{K-1},$$

where $g_t \sim \mathrm{Gumbel}(0, 1)$ and $\tau_e > 0$ is a temperature parameter. The program representation $z = (z_1, \ldots, z_T)$ is then passed to the decoder to execute it on a given input and mapping to the output space. During training, $\tau_e$ is progressively annealed toward zero, tightening the approximation to a true discrete sample and encouraging the model to commit to sparse codebook activations.

### 3.3 Interpreter: Recurrent Neural Program Execution

A common failure point for standard decoders is that they overfit to program lengths and structures seen during training. The neural interpreter, $p_\theta$, uses recurrence to achieve compositional generalisation. This interpreter network conditions on the program representation $z$ one position at a time for a total of $T$ steps, using a shared execution network $d$ to iteratively update an intermediate program state $s_t$. This sequential execution naturally handles novel combinations of primitives and variable program lengths, forcing the model to learn reusable, abstract building blocks. This approach stands in contrast to methods like LPNs, which are limited to representing entire programs in a single monolithic embedding.

The execution process is detailed in Algorithm 1. The input query $x$ is first embedded into an initial hidden state $s_0$ (line 3), and the output distribution is initialised to the one-hot encoding of $x$ (line 4). The state is then refined over $T$ steps. At each step $t$, the execution network takes the previous state $s_{t-1}$ and the embedded latent token $E_v^\top z_t$ to produce output logits $n_t$ (line 6), which are combined with Gumbel noise $h_t$ and passed through a Softmax at temperature $\tau$ to yield a candidate distribution $\tilde{\pi}_t$ over output tokens (line 7). The output distribution is then updated via a skip-gating mechanism (line 8): the probability mass on the skip token $z_{t,\mathrm{skip}}$ linearly interpolates between the previous distribution $\pi_{t-1}$ and the candidate $\tilde{\pi}_t$, allowing the model to effectively ignore a token by placing high probability on the skip. The next state $s_t$ is obtained by projecting the gated distribution $\pi_t$ back through $E_{io}$ (line 9). After all $T$ steps, the final distribution $\pi_T$ is returned as the output (line 10).

---

**Algorithm 1** Interpreter $p_\theta$

---

1: **function** $p_\theta(y \mid x, z, \tau, h)$
2: $\quad (E_v, E_{io}, d) \leftarrow \theta$ $\qquad\qquad$ ▷ code embedder $E_v$, input-output embedder $E_{io}$, execution network $d$
3: $\quad s_0 \leftarrow \mathrm{Embed}(x, E_{io})$ $\qquad\qquad\qquad\qquad\qquad\qquad\qquad\qquad\qquad$ ▷ Embed input query
4: $\quad \pi_0 \leftarrow \mathrm{OneHot}(x)$ $\qquad\qquad\qquad\qquad\qquad$ ▷ Initialise output distribution to input one-hot
5: $\quad$ **for** $t = 1 \to T$ **do**
6: $\qquad n_t \leftarrow d(s_{t-1}, E_v^\top z_t)$ $\qquad\qquad\qquad\qquad$ ▷ Execution network predicts output logits
7: $\qquad \tilde{\pi}_t \leftarrow \mathrm{Softmax}((n_t + h_t)/\tau)$ $\qquad\qquad\qquad$ ▷ Gumbel-Softmax candidate distribution
8: $\qquad \pi_t \leftarrow z_{t,\mathrm{skip}} \cdot \pi_{t-1} + (1 - z_{t,\mathrm{skip}}) \cdot \tilde{\pi}_t$ $\qquad\qquad$ ▷ Skip-gated distribution update
9: $\qquad s_t \leftarrow E_{io}^\top \pi_t$ $\qquad\qquad\qquad\qquad$ ▷ Re-embed gated distribution as next state
10: $\quad$ **return** $\pi_T$ $\qquad\qquad\qquad\qquad\qquad$ ▷ Return final (gated) output distribution

---

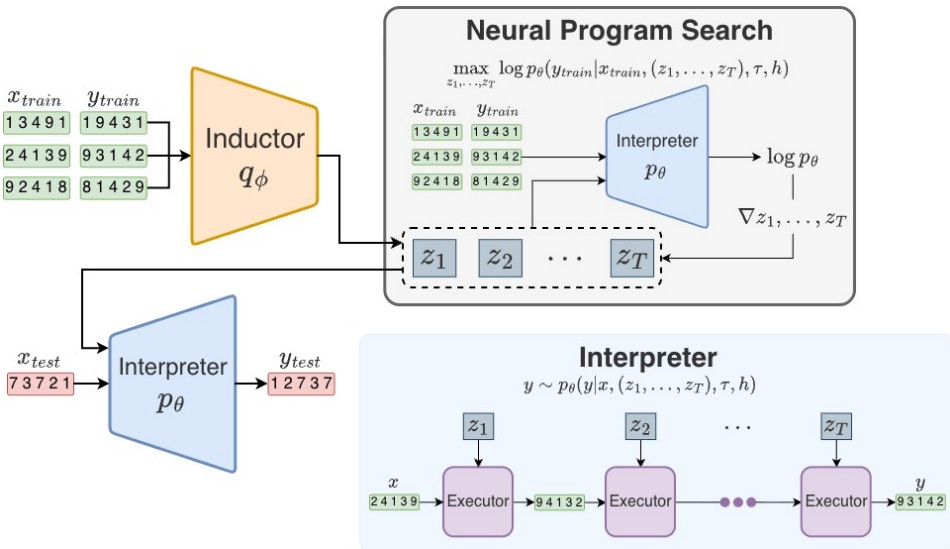

Figure 1: Overview of NLI's inference. The program inductor generates a sequence of latent program tokens conditioned on the specification. Neural Program Search refines this program to better explain the specification, and the neural interpreter then executes the improved program token by token.

### 3.4 SEARCHING NEURAL PROGRAMS

A key benefit of our model is the ability to refine an initial program prediction at test time using gradient-based search. While the encoder provides a fast first guess, it may not be optimal, especially for out-of-distribution programs. Our search operates not in the discrete space of programs, but in a continuous relaxation of the discrete tokens. Because it represents programs as sequences of learned primitives, this space can construct entirely new programs of arbitrary length, even those never seen during training. The search, therefore, becomes a process of discovering these novel compositions.

The key benefit of searching in the relaxed representation of discrete tokens is that, like during training, execution remains fully end-to-end differentiable. This allows us to use gradient ascent to navigate the combinatorial space of possible programs. The search objective is to find the latent embeddings $e^*$ that maximise the expected likelihood of the specification data $S = \{(x_j, y_j)\}$, with the expectation taken over the inductor and interpreter Gumbel samples:

$$e^* = \operatorname*{argmax}_{e} \mathbb{E}_{\substack{g \sim \mathrm{Gumbel}(0,1)^T \\ h \sim \mathrm{Gumbel}(0,1)^T}} \left[ \sum_{(x_j, y_j) \in S} \log p_\theta(y_j \mid x_j, z(e, \tau_e, g), \tau_d, h) \right].$$

Where $z(e, \tau_e, g)$ defines the differentiable mapping from sequence of continuous embeddings $\bar{e}$ to a sequence of categorical distributions over the codebook $z_{1:T}$, defined via the inductor function (Section 3.2). To encourage convergence to a discrete solution, we apply temperature annealing to both $\tau_e$ and $\tau_d$ during optimisation. We begin with a high temperature to smooth the objective and promote broad exploration, and gradually reduce it over time to encourage sharper, more discrete program representations and intermediate computations. At the end of optimisation, the decoder is executed using the final temperature $\tau_d$. While it is possible to sample a fully discrete program before decoding, we instead use the optimised (soft) representation to avoid introducing discrepancies between the optimisation phase and the final execution due to a hard sampling step.

Instead of using a single starting point, we initialise the search from multiple locations to better explore the program space and avoid getting stuck in local minima. Specifically, we sample $s$ initial latent embeddings from a Gaussian distribution, with parameters $\mu = e_t, \sigma = \sigma_s$. We then perform $L$ steps of gradient ascent in parallel from each of the starting points. This search strategy is not only effective for exploration but is also highly scalable, allowing us to use multiple hardware devices.

Most program induction methods have to contend with the problem of finding many programs that explain a given specification. This is a considerable concern in both pure parameter-based modelling and also latent space optimisation (Macfarlane & Bonnet, 2025), where early stopping is often used to limit the chance of converging to programs that do not generalise. In NLI, this is less of a concern as the search space is less flexible than such continuous search spaces, limited to recomposing a set of discrete embeddings. Therefore, the risk of overfitting is relatively limited.

## 4 EXPERIMENTS

We evaluate NLI's compositional generalisation capabilities across a custom benchmark for compositional generalisation and the compositionality version of the DeepCoder benchmark (Balog et al., 2017), introduced in (Shi et al., 2023), comparing to a range of neural and neuro-symbolic baselines.

**Benchmarks**   The custom suite uses fixed-length sequences (20) and is designed to reveal failure modes in the programming-by-example setting, where models see only input-output pairs. It comprises three splits containing different tasks: *Shift-L*, training on small sequence shifts $k \in \{1, \ldots, 5\}$ and testing on larger unseen shifts $k \in \{6, \ldots, 10\}$; *Shift-P*, the inverse, training on large shifts $k \in \{7, 8, 9\}$ and testing on smaller ones $k \in \{1, 2, 3\}$; and *Comp-I*, where models trained on single primitives (e.g., $f(x)$ or $g(x)$) must compose them at test time (e.g., $f(g(x))$). We also explore the compositionality of the DeepCoder dataset, which scales the number of primitives and program complexity. See appendix A for a detailed description of the benchmarks used.

**Baselines**   We compare NLI with In-Context Learning (ICL), Test-Time Training (TTT), Latent Program Networks (LPN), and a discrete variant of LPN (D-LPN), see appendix B for further details. We evaluate three inference strategies for NLI: Base Inference (direct encoder output), Prior Search (sampling from the encoder), and our primary method, Gradient Search, which optimises the program in the latent space. All models are trained with three random seeds, and we report mean performance.

### 4.1 COMPOSITIONAL GENERALISATION IN NEURAL MODELS

Table 1: Performance for different methods and datasets, in the custom suite. We report final accuracy for both in-distribution and out-of-distribution test splits (ID and OOD).

| Method | Shift-L | | Shift-P | | Comp-I | |
| --- | --- | --- | --- | --- | --- | --- |
| | ID | OOD | ID | OOD | ID | OOD |
| In-Context | 1.00 | 0.00 | 1.00 | 0.00 | 1.00 | 0.13 |
| TTT | 1.00 | 0.00 | 1.00 | 0.00 | 0.95 | 0.14 |
| LPN | 1.00 | 0.00 | 1.00 | 0.00 | 1.00 | 0.18 |
| LPN Gradient Search | 1.00 | 0.03 | 1.00 | 0.00 | 1.00 | 0.29 |
| D-LPN | 1.00 | 0.02 | 1.00 | 0.00 | 0.99 | 0.15 |
| D-LPN Gradient Search | 1.00 | 0.01 | 1.00 | 0.00 | 0.99 | 0.20 |
| NLI | 1.00 | 0.00 | 1.00 | 0.00 | 1.00 | 0.17 |
| NLI Prior Search | 1.00 | 0.10 | 1.00 | 0.00 | 1.00 | 0.23 |
| NLI Gradient Search | 1.00 | **0.99** | 1.00 | **1.00** | 1.00 | **0.91** |

We train all models for 100k batches of size 512 and evaluate on held-out test splits, in- and out-of-distribution. Due to the inference cost differences between NLI and baselines, for completeness, for all baselines we also performed training runs with matched compute by increasing decoder layers, for all baselines; this led to a degradation of in-distribution performance and no generalisation. We report the higher, low inference 2-layer decoder results in table 1.

All models achieve near-perfect accuracy on the in-distribution (ID) test sets, demonstrating their ability to solve tasks similar to those seen during training with neural induction. On the more challenging out-of-distribution (OOD) splits, however, all baselines and the non-search variants of our model fail to generalise. In-Context Learning (ICL) and the Latent Program Network (LPN and

```
Learned Program Representations for Shift-L

Ground Truth Program          NLI Program Representation
-----------------------------------------------------
shift_left(1)                 231
shift_left(2)                 231 231
shift_left(3)                 231 231 231
shift_left(4)                 231 476 231
shift_left(5)                 231 476 476
...
shift_left(8)    (OOD)        231 231 231 231 476 476
```

Figure 2: Learned discrete program representations for the Shift-L task. The model composes tokens to represent increasing shift magnitudes, including out-of-distribution generalisation.

D-LPN) show near 0% OOD accuracy on the shift tasks (Shift-L and Shift-P). Search-based LPNs achieve only minor gains on Compose Isolation (Comp-I), but still fail to solve the task. In contrast, NLI with Gradient Search exhibits strong compositional generalisation across all three benchmarks: on Shift-L (length generalisation) it reaches 99% by extrapolating from small to larger unseen shifts; on Comp-I (composing concepts) it achieves 91% by synthesising programs such as $f(g(x))$; and on Shift-P (primitive extraction) it attains a perfect 100% by "decompiling" primitives after training only on complex ones. These results confirm that NLI achieves systematic generalisation, enabled by gradient-based search, whereas the base encoder and prior search variants have performance in line with In-Context, TTT and LPN baselines, which achieve no generalisation. The learned codes reveal systematic reuse of primitives. The model consistently represents a single left shift with token 231 and a two-step shift with token 476, constructing larger programs by combining these two building blocks. The OOD case of eight shifts is also expressed as a mixture of these primitives (found via gradient search), highlighting how generalisation arises from reuse rather than memorisation.

### 4.1.1 Learned Program Representations

We study the task of shifting sequences to the left. During training, the model observes shifts of length 1 to 5 (inclusive). In principle, the network could learn a separate token for each shift. Instead, it discovers a more efficient representation by reusing tokens. Specifically, it learns a token (231) that corresponds to a single left shift. By repeating this token, the network composes shifts of lengths 2 and 3. For larger shifts, it introduces a second token (476) corresponding to a two-step shift. This enables the model to combine primitives to generate more complex shifts. For example, a shift of 4 is represented as one two-step shift plus two one-step shifts. At test time, when generalising OOD to larger shifts, the model composes primitives in the same manner. For instance, to represent an 8-step shift, it uses four single-shift tokens and two two-shift tokens. This demonstrates both compression (a small set of primitives) and compositionality (systematic reuse of primitives).

### 4.2 Understanding the Origins of NLI's Generalisation Capabilities

To investigate the origins of NLI's generalisation, we conduct an ablation study across the datasets in table 1, with results shown in fig. 3. The base model achieves nearly perfect OOD accuracy (97%), and we remove components individually to assess their importance. Most prove indispensable: dropping recurrent execution or the discreteness of either inductor or interpreter representations collapses OOD accuracy to near zero (1–5%). This shows that discrete inductor programs, a discrete interpreter, and recurrent dynamics are all essential for generalisation. Dropping the skip token reduces performance to 24%, consistent with the model's ability to learn

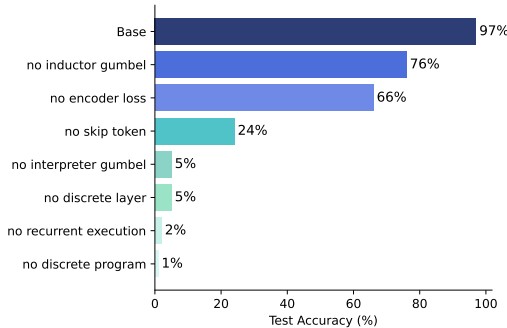

Figure 3: Ablations of the NLI base model to identify components critical for OOD generalisation.

its own skip, but benefits from a dedicated token for faster, more stable training. We also test the importance of the encoder loss, which results in a small drop in performance. The benefit of this loss depends on the type of compositionality that is tested. It is useful for the type of composition in Shift-P, where the program primitives need to be represented to compose new programs that are shorter than those seen during training. For length generalisation, where primitives that have already been seen need to be combined into longer programs, the encoder loss doesn't affect performance.

## 4.3 GUMBEL-SOFTMAX

We observe in fig. 3 that the Gumbel-Softmax relaxation, used to approximate discrete sampling, is a major driver of performance. Removing interpreter-level Gumbel sampling (*no interpreter gumbel*) causes near-complete failure on OOD compositional generalisation (dropping to 5%), while removing inductor-level Gumbel sampling (*no inductor gumbel*) reduces performance from 97% to 76%. Although the encoder still outputs a distribution over a discrete codebook even without Gumbel-Softmax, we conjecture that the network can still learn peaked distributions, allowing discrete representations to emerge and preserving some generalisation. However adding the Gumbel-Softmax approximation strengthens this inductive bias, leading to substantially better performance.

That said, our approach does not fundamentally depend on Gumbel-Softmax; any smooth relaxation of discrete sampling could be substituted (e.g., VQ-VAE with the straight-through estimator (van den Oord et al., 2017)). We chose Gumbel-Softmax primarily for its superior training stability. The straight-through estimator in VQ-VAE is known to suffer from biased gradients, codebook collapse (where many codebook entries remain unused, often requiring oversized codebooks or continual pruning) (Huh et al., 2023), and internal covariate shift between encoder outputs and codebook vectors (Łańcucki et al., 2020). Gumbel-Softmax is not without pitfalls either; stable training requires careful temperature scheduling. We find that annealing the temperature too quickly leads to severe performance degradation, as shown in our ablation on the Shift-L dataset (appendix D). With a gradual annealing schedule, however, training remains reliable and yields the strong results reported.

## 4.4 SCALING TEST-TIME PROGRAM SEARCH

To evaluate the effectiveness of our gradient-based search, we analyse how performance scales with the available computational budget at test time. We benchmark on the Comp-I dataset, varying two key hyperparameters: the number of parallel initialisations (Num starts) and the number of optimisation iterations (Gradient steps). The results, presented in fig. 4, demonstrate a strong and consistent positive correlation between test-time compute and accuracy.

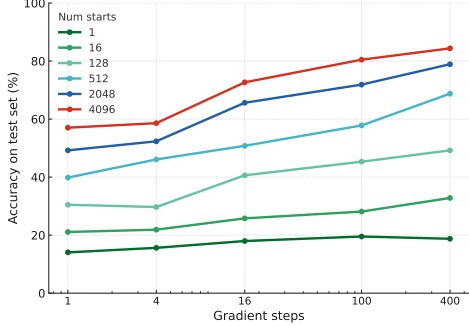

Figure 4: Scaling two test-time axes on Comp-I: gradient steps and number of starts.

## 4.5 DEEPCODER

To assess the scalability of our approach, we evaluate NLI on the DeepCoder benchmark (Shi et al., 2023), a standard testbed for compositional generalisation in program synthesis.

**Dataset overview:** The DeepCoder dataset consists of short functional programs that manipulate lists of integers using a domain-specific language (DSL). Each program is a straight-line sequence of assignments, where every line applies a single operation to either the input or previously defined variables, and the final variable is returned as output.

The DSL includes first-order operations (e.g., `Head`, `Last`, `Take`, `Drop`, `Access`, `Minimum`, `Maximum`, `Reverse`, `Sort`, `Sum`) as well as higher-order functionals (e.g., `Map`, `Filter`,

```
Example Program Execution

Input: [3, 1, 4, 1, 5]

Sort         -> [1, 1, 3, 4, 5]
Map(+1)      -> [2, 2, 4, 5, 6]
Filter(>3)   -> [4, 5, 6]
Sum          -> 15
```

Figure 5: Example of a program written in the DeepCoder DSL.

`Count`, `ZipWith`, `Scan1`), which take a function from a small fixed set of lambda expressions (e.g., `+1`, `*2`, `(-)`, `>0`). We represent programs as sequential transformations of an input,

where each step produces an intermediate result. Figure 5 shows an example, where the arrow (−>) denotes the application of a function to the current sequence.

We note that the original training datasets for DeepCoder composition have not been made public; therefore, data generation was run from scratch to generate datasets of size 11.6 million induction tasks, to train the neural baselines (NLI, LPN and In-Context). Due to the prohibitive costs of the data sampling function, this is less than the 60 million used in the original work; however, baselines all achieve competitive performance. We highlight that neuro-symbolic approaches all leverage access to ground-truth programs during training, where NLI and LPN do not require this. However, adding natural language program representations during training can serve as a powerful training signal for the neural decoders, which are otherwise bottlenecked by the encoder's induction capacity. Therefore we add NLI w/ program and LPN w/ program baselines. These leverage an additional encoder mapping from program representations to latent space, resulting in an additional reconstruction loss, which is simply added to the total loss with equal weight to the standard encoder reconstruction loss. We give a complete description of the natural language program encoder and our training procedure in appendix E. In contrast, NLI, along with neural baselines such as LPN, and an In-Context baseline, must induce program behaviour solely from input-output pairs. All neural benchmarks were trained for 200k batches of size 512, see appendix C for more details. We find that end-to-end neural

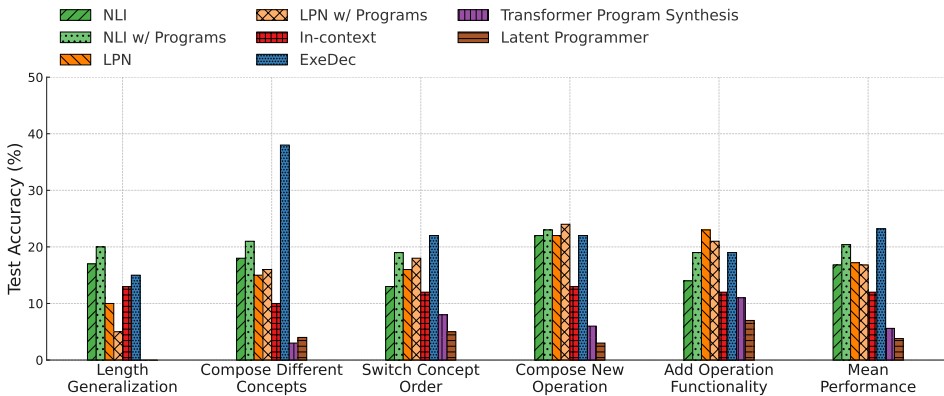

Figure 6: Comparison of fully neural baselines and NLI against neuro-symbolic methods. Neuro-symbolic models (ExeDec, Transformer, Latent Programmer) use ground-truth program annotations, while neural models (In-Context, LPN, NLI) rely only on input–output pairs.

methods such as NLI and LPN substantially outperform earlier Latent Programmer approaches and Transformer-based program synthesis. Secondly, despite the absence of program supervision, they achieve performance competitive with ExeDec (Shi et al., 2023), highlighting the capacity of neural programming-by-example approaches to autonomously discover structured program representations.

A direct comparison between NLI and LPN further reveals complementary strengths. NLI generalises more effectively to longer programs and novel concept compositions, which is a strength of NLI due to its ability to compose programs of arbitrary length. In contrast, LPN excels at switching concept order and extending functionality with new operations. These differences suggest that their learned latent structures capture distinct inductive biases, leading to different generalisation behaviours.

## 5 RELATED WORK

**Symbolic Program Synthesis.** Early work in program synthesis largely relied on symbolic techniques and DSLs. Classical systems, such as those by Summers (1977) and Gulwani (2011), used predefined DSLs with explicit search over symbolic programs. These methods provide interpretability and exactness but suffer from scalability issues, as every new domain requires manual DSL design. Recent neuro-symbolic hybrids, such as DeepCoder (Balog et al., 2017), combine a neural predictor with symbolic search, predicting program components to accelerate search. However, their reliance on restricted DSLs limits generalisation beyond the designed primitives.

**Neural Program Induction and Meta Learning** Neural approaches aim to overcome the brittleness of symbolic methods by learning programs directly from examples. Neural Programmer-Interpreters (Reed & De Freitas, 2015) execute programs implicitly with recurrent models, while Devlin et al. (2017) introduced meta-induction for few-shot learning. These models improve adaptability but often fail to generalise compositionally and demand large supervision. ExeDec (Shi et al., 2023) added execution decomposition as an inductive bias, yet still relies on ground-truth decompositions and remains costly. Meta learning advances this by training networks to adapt across task distributions (Finn et al., 2017), a setup closely related to the optimisation considered here.

**Latent Representations of Programs.** Another line of work introduces latent spaces to represent programs more flexibly. CompILE (Kipf et al., 2019) segments demonstrations into reusable latent codes with Gumbel-Softmax relaxation, showing benefits for imitation learning. The Latent Programmer (Hong et al., 2021) extends this idea to discrete latent codes that plan over input-output examples, using a VQ-VAE style autoencoder with beam search in latent space, see appendix F for a discussion on its relation to NLI. Most recently, Latent Program Networks (LPNs) (Macfarlane & Bonnet, 2025) proposed continuous latent program representations to facilitate test-time search, but the lack of discrete compositional structure hinders combinatorial generalisation.

**Compositionality and Generalisation.** Compositional generalisation remains a central challenge in neural program synthesis. Lake & Baroni (2018) demonstrated that standard seq2seq models fail to generalise systematically to novel compositions. Approaches such as the Compositional Recursive Learner (CRL) (Chang et al., 2019) attempt to address this by learning to compose reusable transformations. Similarly, recursion-based methods (Cai et al., 2017) leverage inductive biases from programming languages to handle inputs of greater complexity than those seen during training. While these directions highlight the importance of compositional structure, they either rely on strong supervision or achieve only limited scalability.

**Discrete Representation Learning.** Discrete latent representations provide a natural way to capture compositional structure and improve interpretability. The Vector-Quantized Variational Autoencoder (VQ-VAE) (van den Oord et al., 2017) exemplifies this approach by learning a finite codebook of tokens, with gradients passed via a straight-through estimator. A complementary method is the Gumbel-Softmax relaxation (Jang et al., 2017; Maddison et al., 2017), which reparameterises categorical sampling with a differentiable approximation. Together, these techniques enable end-to-end training with discrete variables while retaining symbolic structure. A practical example arises in hierarchical reinforcement learning, where the options framework (Sutton et al., 1999) defines a set of reusable, temporally extended actions that compose into complex behaviours.

## 6 CONCLUSION

In this work, we introduced the Neural Language Interpreter, a novel architecture that bridges the divide between symbolic and neural approaches in program synthesis. By learning a discrete, symbolic-like language and a differentiable interpreter, NLI combines the compositional strengths of symbolic systems with the flexibility of neural networks. We show that NLI outperforms existing neural methods and is competitive with symbolic ones on challenging compositional generalisation tasks, with ablations confirming that the discrete, sequential latent program is key to this success.

**Limitations and Future Work:** NLI introduces a new paradigm for program induction that demonstrates promising compositional generalisation. While we believe it has strong potential to scale to harder problems, the present work is an initial exploration and comes with several limitations. A primary bottleneck is the computational cost of test-time search in the latent representation space; although NLI proves remarkably robust to overfitting even under constrained search budgets, scaling to more complex tasks will likely require more efficient inference strategies, with evolutionary/local search being promising directions. As problem difficulty increases, programs will grow both in length and vocabulary size, potentially leading to vanishing or exploding gradients, which, while not observed in our experiments, could require architectural modifications at scale. The current interpreter also limits expressive power: each layer conditions on exactly one token, preventing parameterised primitives (e.g., `add(k)` for variable $k$), and execution follows a strictly sequential flow without conditional branching.

ACKNOWLEDGMENTS

We thank François Chollet for early inspirational conversations about the importance of developing a neural program that could handle recurrence and length generalisation by design, which gave us inspiration for the final architecture. This research was supported by Canada's NSERC and the CIFAR AI Chairs program. Matthew Macfarlane is supported by LIFT project 019.011, which is partly financed by the Dutch Research Council (NWO). This research was supported by Cloud TPUs from Google's TPU Research Cloud (TRC) and by GPU credits provided by Modal. We also thank the anonymous reviewers for their comments and helpful discussions that improved the final version of the paper.

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

# A DATASETS

## A.1 COMPOSITIONALITY BENCHMARK

We constructed the Compositionality Benchmark using our own sampling and problem synthesis procedures to evaluate distinct facets of compositional reasoning. The benchmark comprises three main tasks designed to probe different dimensions of generalisation. For each task, dataset sizes were chosen to provide a robust training scale and sufficient evaluation coverage. The three splits are:

1. **Permutation Length Generalisation (`Shift-L`).** This task measures extrapolation on a parameterised function. The model learns a `left_shift(n)` operation on a sequence. Training is restricted to a small, contiguous range of integer shifts, specifically for $n \in \{1, 2, 3, 4, 5\}$. Evaluation is performed on larger, unseen shift values, $n \in \{6, 7, 8, 9, 10\}$. This measures the model's ability to generalise beyond the magnitude of parameters observed during training.

2. **Sub-Function Extraction (`Shift-P`).** This task tests whether a model can extract reusable primitives from programs seen only at longer lengths. The underlying operation is again `left_shift(n)`. Training is performed on (e.g., $n \in \{7, 8, 9\}$). Evaluation then probes generalisation to a different, unseen range of values (e.g., $n \in \{1, 2, 3\}$), testing whether the model has learned the abstract concept of "shifting by n" rather than memorising separate programs for each training example.

3. **Composition of Primitives (`Comp-I`).** This task evaluates whether a model can compose primitive functions it has only seen in isolation. The model is provided with a library of over 20 primitive sequence-to-sequence operations (e.g., `reverse`, `shift_left_3`, `increment_2`). During training, the model only sees programs consisting of a single primitive operation. For evaluation, it must execute programs that are compositions of two or more primitives, testing for generalisation from individual operations to novel compositions.

Table 2: Dataset sizes for the Compositionality Benchmark.

| Split | Size |
|---|---|
| Train | 2,000,000 |
| Test | 10,000 |

## A.2 DEEPCODER

We use the DeepCoder domain and adopt the compositional generalisation splits from the ExeDec codebase (Shi et al., 2023). Following their Domain-Specific Language (DSL) and splitting procedures, we sampled 2,000,000 training tasks and 10,000 test tasks. The five splits are designed to probe different dimensions of compositional generalisation:

1. **Length-Generalisation.** Training programs contain 1–4 lines, while test programs have length 5. This evaluates whether models can extrapolate to deeper compositions than observed during training (Balog et al., 2017).

2. **Compose-Different-Concepts.** Operations are partitioned into two groups: (i) all first-order operations plus `Map`, and (ii) all remaining higher-order operations. Training only composes within a single group, while test programs require mixing across groups. This measures cross-concept compositionality.

3. **Switch-Concept-Order.** Training tasks always compose operations in a fixed group ordering (e.g., first-order → higher-order), while test tasks reverse the ordering. This evaluates whether models can generalise to new sequential structures of concepts.

4. **Compose-New-Operation.** The held-out operation is `Scanl1`. Training tasks either use `Scanl1` in isolation (25% of tasks) or exclude it entirely, while test tasks require `Scanl1` to be composed with other operations. This probes whether the model can generalise an operator from isolated usage to composed contexts.

5. **Add-Operation-Functionality.** Training only uses `Scanl1` with lambdas $(-)$ and `min`. Test tasks require `Scanl1` with new lambdas $(+), (\times)$, and `max`. This tests whether models can extend their understanding of a known operator by analogy to other operations.

Table 3: Dataset sizes for DeepCoder, generated using the ExeDec repository.

| Split | Size |
|---|---|
| Train | 11,600,000 |
| Test | 10,000 |

# B  BASELINES

## B.1  D-LPN

Discrete Latent Program Networks (D-LPN) is an augmented version of Latent Program Networks in which the continuous latent representation is replaced with a discrete bottleneck, represented as a sequence of tokens. Given a specification $S = \{(x_i, y_i)\}$, an encoder $q_\phi$ maps $S$ to a sequence of discrete latent variables, trained using a Gumbel-Softmax relaxation to enable backpropagation. This sequence of discrete tokens is then prepended to the decoder input sequence, and the decoder $p_\theta$ predicts outputs for a given query input. The primary purpose of this baseline is to disentangle the effect of a discrete latent space from the recurrent execution mechanism that enables compositionality in NLI.

## B.2  IN-CONTEXT

The in-context learning baseline treats program induction as a conditional sequence modelling problem. A transformer-based model is trained to predict outputs given a concatenated context consisting of input–output pairs $(x_i, y_i)$ and a query input $x_{n+1}$. The model directly produces a prediction $y_{n+1} \sim p_\theta(\cdot \mid S, x_{n+1})$, where all adaptation is performed implicitly within the forward pass. In this setting, no explicit latent program representation is constructed, and the model parameters remain fixed at test time.

## B.3  TTT: TEST-TIME TRAINING

Starting from pretrained parameters $\theta$, TTT constructs a self-supervised loss from the specification itself: for each $(x_i, y_i) \in S$, the model predicts $y_i$ given $x_i$ and the remaining examples $S \setminus \{(x_i, y_i)\}$, matching the in-context conditioning scheme used during pretraining. A small number of gradient steps on this loss yield adapted parameters $\theta'$. Predictions for a new query $x_{n+1}$ are then made by conditioning the updated model on the full specification, $y_{n+1} \sim p_{\theta'}(\cdot \mid S, x_{n+1})$.

## C  HYPERPARAMETERS

Table 4: Model Hyperparameters for NLI.

| Hyperparameter | Shift-L | Shift-P | Compose I | DeepCoder |
|---|---|---|---|---|
| **Model Architecture** | | | | |
| Model Dimension ($d_{\text{model}}$) | 128 | 128 | 128 | 128 |
| Number of Heads ($n_{\text{head}}$) | 8 | 8 | 8 | 8 |
| Feed-Forward Dimension ($d_{\text{ff}}$) | 512 | 512 | 512 | 512 |
| Encoder Layers | 2 | 2 | 2 | 4 |
| Decoder Layers | 2 | 2 | 2 | 2 |
| Positional Embedding | Sinusoidal | Sinusoidal | Sinusoidal | Sinusoidal |
| Gradient Clip Norm | 2.0 | 2.0 | 2.0 | 2.0 |
| **Program Generation** | | | | |
| Program Vocabulary Size | 512 | 512 | 512 | 512 |
| Program Length (Training) | 10 | 10 | 4 | 4 |
| **Training** | | | | |
| Learning Rate | 2e-4 | 2e-4 | 2e-4 | 2e-4 |
| Num Batches | 100k | 100k | 100k | 200k |
| **Gumbel-Softmax Sampling (Inductor)** | | | | |
| Use Inductor Gumbel | True | True | True | True |
| Start Temperature | 8.0 | 8.0 | 8.0 | 8.0 |
| End Temperature | 0.5 | 0.5 | 0.5 | 0.5 |
| Annealing Batches | 20,000 | 20,000 | 100,000 | 200,000 |
| Decay Strategy | Exponential | Exponential | Exponential | Exponential |
| Straight-Through | False | False | False | False |
| **Gumbel-Softmax Sampling (Interpreter)** | | | | |
| Use Interpreter Gumbel | True | True | True | True |
| Start Temperature | 2.0 | 2.0 | 2.0 | 2.0 |
| End Temperature | 0.5 | 0.5 | 0.5 | 0.5 |
| Annealing Batches | 20,000 | 20,000 | 100,000 | 200,000 |
| Decay Strategy | Exponential | Exponential | Exponential | Exponential |
| Straight-Through | False | False | False | False |
| **Regularization & Losses** | | | | |
| Encoder Loss Coefficient | 1e-5 | 1e-5 | 1e-5 | 1e-5 |
| **Search** | | | | |
| Gradient Steps | 100 | 100 | 100 | 100 |
| Number of Initializations | 1024 | 1024 | 8192 | 1024 |
| Std for initialisation ($\sigma_s$) | 7.5 | 7.5 | 7.5 | 7.5 |

Table 5: Model Hyperparameters for LPN. The same default configuration was used across all datasets.

| Hyperparameter | Shift-L | Shift-P | Compose I | DeepCoder |
|---|---|---|---|---|
| **Model Architecture** | | | | |
| Model Dimension ($d_{\text{model}}$) | 512 | 512 | 512 | 512 |
| Number of Heads ($n_{\text{head}}$) | 8 | 8 | 8 | 8 |
| Feed-Forward Dimension ($d_{\text{ff}}$) | 512 | 512 | 512 | 512 |
| Encoder Layers | 2 | 2 | 2 | 4 |
| Decoder Layers | 2 | 2 | 2 | 2 |
| Use Layer normalisation | True | True | True | True |
| Positional Embedding | Sinusoidal | Sinusoidal | Sinusoidal | Sinusoidal |
| Dropout Rate | 0.0 | 0.0 | 0.0 | 0.0 |
| VAE Beta ($\beta$) | 1e-3 | 1e-3 | 1e-3 | 1e-3 |
| Gradient Clip Norm | 2.0 | 2.0 | 2.0 | 2.0 |
| **Training** | | | | |
| Learning Rate | 2e-4 | 2e-4 | 2e-4 | 2e-4 |
| Num Batches | 100k | 100k | 100k | 200k |

Table 6: Model Hyperparameters for D-LPN. The same default configuration was used across all datasets.

| Hyperparameter | Shift-L | Shift-P | Compose I |
|---|---|---|---|
| **Model Architecture** | | | |
| Model Dimension ($d_{\text{model}}$) | 512 | 512 | 512 |
| Number of Heads ($n_{\text{head}}$) | 8 | 8 | 8 |
| Feed-Forward Dimension ($d_{\text{ff}}$) | 512 | 512 | 512 |
| Encoder Layers | 2 | 2 | 2 |
| Decoder Layers | 2 | 2 | 2 |
| Use Layer Normalisation | True | True | True |
| Positional Embedding | Sinusoidal | Sinusoidal | Sinusoidal |
| Dropout Rate | 0.0 | 0.0 | 0.0 |
| Gradient Clip Norm | 2.0 | 2.0 | 2.0 |
| **Training** | | | |
| Learning Rate | 2e-4 | 2e-4 | 2e-4 |
| Num Batches | 100k | 100k | 100k |
| **Gumbel-Softmax Sampling (Inductor)** | | | |
| Use Inductor Gumbel | True | True | True |
| Start Temperature | 8.0 | 8.0 | 8.0 |
| End Temperature | 0.5 | 0.5 | 0.5 |
| Annealing Batches | 20,000 | 20,000 | 100,000 |
| Decay Strategy | Exponential | Exponential | Exponential |
| Straight-Through | False | False | False |
| **Gumbel-Softmax Sampling (Interpreter)** | | | |
| Use Interpreter Gumbel | True | True | True |
| Start Temperature | 2.0 | 2.0 | 2.0 |
| End Temperature | 0.5 | 0.5 | 0.5 |
| Annealing Batches | 20,000 | 20,000 | 100,000 |
| Decay Strategy | Exponential | Exponential | Exponential |
| Straight-Through | False | False | False |

Table 7: Model Hyperparameters for In-context. The same default configuration was used across all datasets.

| Hyperparameter | Shift-L | Shift-P | Compose I | DeepCoder |
|---|---|---|---|---|
| **Model Architecture** | | | | |
| Model Dimension ($d_{\text{model}}$) | 512 | 512 | 512 | 512 |
| Number of Heads ($n_{\text{head}}$) | 8 | 8 | 8 | 8 |
| Feed-Forward Dimension ($d_{\text{ff}}$) | 512 | 512 | 512 | 512 |
| Encoder Layers | 2 | 2 | 2 | 4 |
| Decoder Layers | 2 | 2 | 2 | 2 |
| Use Layer normalisation | True | True | True | True |
| Positional Embedding | Sinusoidal | Sinusoidal | Sinusoidal | Sinusoidal |
| Dropout Rate | 0.0 | 0.0 | 0.0 | 0.0 |
| Gradient Clip Norm | 2.0 | 2.0 | 2.0 | 2.0 |
| **Training** | | | | |
| Learning Rate | 2e-4 | 2e-4 | 2e-4 | 2e-4 |
| Num Batches | 100k | 100k | 100k | 200k |

## D  GUMBEL-SOFTMAX TEMPERATURE ANNEALING ABLATION

In this section, we investigate how the base NLI model learns discrete program representations under different Gumbel-Softmax temperature annealing schedules. Stable training requires careful control of the program- and layer-level temperatures, and here we ablate the effect of the shared annealing duration on the Shift-L dataset.

Our model uses two independent Gumbel-Softmax temperatures:

- **Inductor temperature** ($\tau_{\text{prog}}$): controls the discreteness of the tokenised high-level program.
- **Interpreter temperature** ($\tau_{\text{layer}}$): controls the discreteness of per-token execution choices within each layer.

Both temperatures are linearly annealed over the same number of steps ($\tau_{\text{prog}}$: 8.0→0.5, $\tau_{\text{layer}}$: 2.0→0.5), after which they are held fixed at 0.5 for the remainder of training. All runs use 100k total training steps and identical hyperparameters, varying only the shared annealing duration.

Table 8: Ablation of the shared Gumbel-Softmax temperature annealing duration on Shift-L (in-distribution accuracy; 100k total training steps, averaged over 3 seeds).

| Annealing duration | NLI (ID) |
|---|---|
| 1k steps | 0.00 |
| 5k steps | 0.41 |
| 10k steps | 1.00 |
| 20k steps | 1.00 |
| 50k steps | 1.00 |
| 100k steps | 1.00 |

Short annealing schedules (1k–5k steps) fail to produce stable discrete program representations. Accuracy increases sharply at 10k steps, after which all longer schedules perform identically. This shows that the model is robust to the exact duration once a minimal threshold is reached, and that our default 20k-step schedule lies well within the stable regime for Shift-L.

# E    NATURAL LANGUAGE PROGRAM ENCODER

In this section, we describe the natural language program encoder used alongside the input–output (I/O) encoder. The natural language program encoder can be leveraged when data is available to provide a stronger, more direct training signal to the neural executor. When training relies solely on the I/O encoder, the discrete program latents can become bottlenecked by the encoder's limited inductive capacity, particularly on harder tasks. Providing ground-truth programs during training alleviates this issue and allows the executor to learn the correct program primitives more effectively. Below, we outline the tokenisation scheme, architecture, and training procedure.

## E.1    TOKENISATION

Programs are whitespace-tokenised according to the DeepCoder DSL. The full vocabulary contains 153 tokens: 4 special tokens (PAD, <BOS>, <EOS>, |), 4 structural tokens (=, INPUT, [, ]), 15 operations, 19 lambda functions, 10 variable tokens x0–x9, and integer literals from $-50$ to $50$.

**Example**

String representation: `x0 = INPUT | x1 = Map (+1) x0 | x2 = Filter (>0) x1 | x3 = Head x2`

Token sequence (with <BOS> and <EOS>): `<BOS> x0 = INPUT | x1 = Map (+1) x0 | x2 = Filter (>0) x1 | x3 = Head x2 <EOS>`

Tokenised sequence (token IDs): `1 42 4 5 3 43 18 23 42 3 44 19 33 43 3 45 8 44 2`

## E.2    ARCHITECTURE

The natural language program encoder uses the same architecture and hyperparameters as the I/O encoder in all experiments. Both encoders share the same latent codebook for representing discrete latent programs and the Gumbel–Softmax relaxation to learn these representations.

## E.3    TRAINING WITH THE PROGRAM ENCODER

Access to ground-truth programs $P$ allows the model to bypass the inductive bottleneck of the I/O encoder and directly expose the executor to correct program structures. We do not introduce a separate program decoder; instead, both encoders share the neural execution decoder $p_\theta$.

During training, the natural language program encoder maps each tokenised program $P$ to discrete latents using the shared codebook. The total objective adds an auxiliary reconstruction term weighted by $\lambda_{\text{prog}}$, which we set to $1.0$ in all experiments.

$$\mathcal{L}_{\text{rec}} = \mathcal{L}_{\text{IO\_rec}} + \lambda_{\text{prog\_rec}} \, \mathcal{L}_{\text{prog\_rec}}$$

To ensure both encoders learn a unified latent space, we control gradient flow as follows. Gradients from $\mathcal{L}_{\text{prog\_rec}}$ update both the program encoder parameters and the executor, whereas gradients from $\mathcal{L}_{\text{IO\_rec}}$ do not update the executor (stop-gradient), forcing the I/O encoder to align with the program encoder's higher-quality latents.

At test time, the program encoder is discarded. Only the trained I/O encoder and executor are used for inference and search.

## F    COMPARISON TO DISCRETE LATENT PROGRAMMER

We compare our model with the Discrete Latent Programmer (DLP) (Hong et al., 2021), which also employs discrete latent codes for program induction. While both approaches share this high-level similarity, they diverge substantially in their architectures, training assumptions, and mechanisms for test-time adaptation. The key differences lie in how programs are executed and how search is performed at inference.

### F.1    PROGRAM EXECUTION AND SUPERVISION

In our model, program tokens are interpreted by a recurrent neural interpreter that applies each token as an operation to an intermediate state. This sequential execution enables variable-length programs, promotes compositional reuse of learned primitives, and allows the model to be trained end-to-end on raw input–output examples alone. Since outputs can be directly compared to targets, no ground-truth program annotations are required.

DLP, by contrast, does not include a neural interpreter. Its decoder predicts full program sequences from latent codes, and training requires access to the underlying program representations. This reliance on program supervision restricts DLP to domains where the generating programs are known and a domain-specific language is available, limiting its applicability beyond synthetic benchmarks.

### F.2    TEST-TIME PROGRAM SEARCH

A further distinction arises in test-time adaptation. Our model exploits the differentiability of the neural interpreter to refine latent program embeddings via gradient-based search. This procedure enables efficient adaptation: initial program guesses from the encoder can be continuously optimised to better fit new examples, even when they require novel compositions not seen during training.

In contrast, DLP performs beam search in the discrete program space. This search is combinatorial, lacks gradient guidance, and cannot refine programs based on execution error. As a result, DLP's generalisation is hindered, particularly in out-of-distribution settings where small corrections to a predicted program are necessary. By enabling gradient-based refinement in a relaxed latent space, our model provides a more powerful and adaptive mechanism for program synthesis.

# G  NLI Program Representations

In this section, we provide examples of the discrete latent codes discovered by NLI, both for in-distribution programs and for how these primitives are composed by search to generalise out-of-distribution (OOD).

## G.1  Shift-L

We study the task of shifting sequences to the left. During training, the model observes shifts of length 1 to 5 (inclusive). In principle, the network could learn a separate token for each shift. Instead, it discovers a more efficient representation by reusing tokens. Specifically, it learns a token (231) that corresponds to a single left shift. By repeating this token, the network composes shifts of lengths 2 and 3. For larger shifts, it introduces a second token (476), which corresponds to a two-step shift. This enables the model to combine primitives to generate more complex shifts.

For example, a shift of 4 is represented as one two-step shift plus two one-step shifts. At test time, when generalising OOD to larger shifts, the model composes primitives in the same manner. For instance, to represent an 8-step shift, it uses four single-shift tokens and two two-shift tokens. This demonstrates both compression (a small set of primitives) and compositionality (systematic reuse of primitives).

```
--------------------------------------------------------------------

Example 1: Shift Left by 1

Task Specification:
Input:  [8, 2, 5, 9, 1, 6, 3, 4, 7, 0]
Output: [2, 5, 9, 1, 6, 3, 4, 7, 0, 8]

Ground Truth Program: y = left_shift(x, 1)
NLI Program Representation: 231

--------------------------------------------------------------------

Example 2: Shift Left by 2

Task Specification:
Input:  [4, 6, 7, 1, 9, 0, 3, 8, 5, 2]
Output: [7, 1, 9, 0, 3, 8, 5, 2, 4, 6]

Ground Truth Program: y = left_shift(x, 2)
NLI Program Representation: 231 231

--------------------------------------------------------------------

Example 3: Shift Left by 3

Task Specification:
Input:  [3, 7, 4, 0, 6, 2, 9, 5, 8, 1]
Output: [0, 6, 2, 9, 5, 8, 1, 3, 7, 4]

Ground Truth Program: y = left_shift(x, 3)
NLI Program Representation: 231 231 231

--------------------------------------------------------------------

Example 4: Shift Left by 4

Task Specification:
Input:  [5, 1, 9, 2, 8, 6, 0, 7, 3, 4]
Output: [8, 6, 0, 7, 3, 4, 5, 1, 9, 2]

Ground Truth Program: y = left_shift(x, 4)
```

```
43  NLI Program Representation: 231 476 231
44
45  --------------------------------------------------------------------
46
47  Example 5: Shift Left by 5
48
49  Task Specification:
50  Input:  [9, 4, 1, 5, 2, 7, 6, 0, 3, 8]
51  Output: [7, 6, 0, 3, 8, 9, 4, 1, 5, 2]
52
53  Ground Truth Program: y = left_shift(x, 5)
54  NLI Program Representation: 231 476 476
55
56  --------------------------------------------------------------------
57
58  Example 6 (OOD): Shift Left by 8
59
60  Task Specification:
61  Input:  [0, 1, 2, 3, 4, 5, 6, 7, 8, 9]
62  Output: [8, 9, 0, 1, 2, 3, 4, 5, 6, 7]
63
64  Ground Truth Program: y = left_shift(x, 8)
65  NLI Program Representation: 231 231 231 231 476 476
66
67  --------------------------------------------------------------------
```

Listing 1: Learned NLI Program Representations for List Shift Tasks.

## H  APPENDIX: TOKEN REUSE REGULARISATION LOSS

To bias the model toward a compact, reusable set of primitives, we add a regulariser to the encoder's output distribution. It penalises the expected number of distinct program tokens appearing in a batch, under a simplifying independence assumption that makes the expectation differentiable and directly optimisable by gradient descent. Encouraging reuse pushes the encoder to discover a small set of operations that recombine to solve many tasks, supporting compositional generalisation.

Let $P$ denote the tensor of token probabilities output by the encoder, with dimensions $(B, N, V)$, where $B$ is the batch size, $N$ the program length, and $V$ the vocabulary size. We write $p_{b,i,k}$ for the probability of token $k$ at position $i$ in sequence $b$.

We treat the event "token $k$ is selected at position $(b, i)$" as an independent Bernoulli trial with parameter $p_{b,i,k}$, ignoring the within-position anti-correlation induced by the softmax. Under this approximation, the probability that token $k$ never appears in the batch is

$$\mathbb{P}(\text{token } k \text{ never appears}) = \prod_{b=1}^{B} \prod_{i=1}^{N} (1 - p_{b,i,k}), \tag{4}$$

which we compute in log-space for numerical stability,

$$\log \mathbb{P}(\text{token } k \text{ never appears}) = \sum_{b=1}^{B} \sum_{i=1}^{N} \log(1 - p_{b,i,k}). \tag{5}$$

The probability that token $k$ appears at least once is then

$$\mathbb{P}(\text{token } k \text{ appears}) = 1 - \exp\left( \sum_{b=1}^{B} \sum_{i=1}^{N} \log(1 - p_{b,i,k}) \right), \tag{6}$$

and by linearity of expectation the approximate expected number of distinct tokens used in the batch is

$$\mathcal{L}_{\text{reg}} = \sum_{k=1}^{V} \mathbb{P}(\text{token } k \text{ appears}). \tag{7}$$

### PRACTICAL CONSIDERATIONS

The token reuse loss is added to the encoder loss with weight $\lambda_{\text{reg}}$. As a batch-level statistic it can be sensitive to batch size, but with the large batches we use in practice we did not observe instabilities. Note that $\mathcal{L}_{\text{reg}} \in [0, V]$, so $\lambda_{\text{reg}}$ must be tuned relative to the vocabulary size. The loss is trivially minimised by collapsing to a single-token language, so the regulariser cannot prevent collapse on its own. In practice we use a small $\lambda_{\text{reg}}$ so that the encoder loss dominates, and the regulariser acts as a gentle inductive bias toward compact vocabularies rather than a hard constraint.

# I  THE USE OF LARGE LANGUAGE MODELS (LLMS)

In this work, large language models (LLMs) were used solely as a tool for polishing the writing, specifically to remove grammatical and spelling errors. They did not contribute to research ideation or any other significant aspects of the paper.

