# OpenReview forum: "Gradient-Based Program Synthesis with Neurally Interpreted Languages"
_ICLR.cc/2026/Conference — ICLR 2026 Poster_

### Official Review · Reviewer_xGro · 2025-10-29

**Soundness:** 3
**Presentation:** 3
**Contribution:** 2
**Rating:** 6
**Confidence:** 4

**Summary:**

This paper introduces the Neural Language Interpreter (NLI), a novel architecture that learns a discrete, symbolic-like programming language and a differentiable interpreter end-to-end. The model is evaluated on several compositional generalization benchmarks and demonstrates strong out-of-distribution performance, particularly when combined with gradient-based search at test time.

**Strengths:**

- Neuron-symbolic is an important research area
- The model shows impressive compositional generalization results
- Using a neural executor is a nice idea, which is similar to a world model

**Weaknesses:**

- Gumbel-softmax has been applied to many research areas to publish papers. However, it seems that no influential results have been produced by Gumbel-softmax. I doubt the impact of this paper, too.

**Questions:**

Are there any influential applications of Gumbel-softmax in any area? Could you give me some examples?

---

> ### Author Response · Authors · 2025-11-21
>
> We thank the reviewer for their positive remarks and for highlighting the strengths of our work, including its contributions to neurosymbolic methods and compositional generalisation. We also appreciate the connection you draw between our neural executor and world models, and we have included it in the final version.
>
> Below, we directly address your concern regarding the use of Gumbel-Softmax. We have included a discussion on the use of Gumbel softmax, its impact on performance and the importance of temperature annealing for stable training in the main body of the paper in Section 5.3, with further details in Appendix C.
>
> A core enabler of compositional generalisation in our Neural Language Interpreter (NLI) is approximate discrete sampling. This enables learning composable discrete representations, which allow end-to-end training via gradient descent while enabling compositionality at test time, capabilities that neural networks have struggled to achieve. Importantly, our method does not rely specifically on the Gumbel-Softmax approximation for learning such discrete representations; any smooth approximation of discrete sampling could be substituted, such as VQ-VAE with the straight-through estimator [1]. If a superior alternative emerges, it could be integrated into our architecture without methodological changes.
>
> That said, we chose Gumbel-Softmax over VQ-VAE for practical reasons related to training stability. The straight-through estimator in VQ-VAE can introduce challenges, including biased gradients and codebook collapse (where specific tokens go unused, necessitating large codebooks or ongoing token pruning that adds complexity) [2]. These weaknesses are well-documented in prior work, which highlights that VQ-VAE training can suffer from instability, under-utilisation of codebooks, and internal covariate shift between encoder outputs and codewords [7]. Given the novelty of our approach, we prioritised the simpler, more stable Gumbel-Softmax, which also does not require additional losses that VQ-VAE requires.
>
>
> To address your question on influential applications of Gumbel-Softmax, we highlight several high-impact works across key areas:
>
> - **Neural Architecture Search**: FBNet [3] and SNAS [4] leverage Gumbel-Softmax for differentiable search, enabling hardware-aware efficient ConvNet design and stochastic path sampling that outperform prior methods in speed and performance.
> - **Image Generation**: The DALL·E model [5] uses Gumbel-Softmax for zero-shot text-to-image generation, achieving state-of-the-art results on benchmarks like MS-COCO.
> - **Sequence Generation**: GANs for Sequences of Discrete Elements [6] introduces Gumbel-Softmax to enable end-to-end training of GANs on discrete data, opening avenues for text and sequence modelling.
>
> While these examples underscore Gumbel-Softmax as a principled tool for differentiable discrete sampling, we believe that this paper itself, given the strong generalisation results, could pave a path forward for increased usage of discrete sampling approximations going forward, with the difference between discrete and continuous striking, 0% vs 99-100% on Shift-L and Shift-P datasets. Notably, our ablation studies demonstrate that layer-level Gumbel-Softmax is essential for out-of-distribution (OOD) generalisation in NLI, yielding significant gains over baselines without it.
>
> We hope this clarifies our choices and strengthens the case for the work's contributions.
>
>
>
> ### References
> [1] A. van den Oord, O. Vinyals, and K. Kavukcuoglu, "Neural Discrete Representation Learning," in *Advances in Neural Information Processing Systems (NeurIPS)*, 2017.
> [2] M. Huh, B. Cheung, P. Agrawal, and P. Isola, "Straightening Out the Straight-Through Estimator: Overcoming Optimization Challenges in Vector Quantized Networks," in *International Conference on Machine Learning (ICML)*, 2023.
> [3] B. Wu, et al., "FBNet: Hardware-Aware Efficient ConvNet Design via Differentiable Neural Architecture Search," in *IEEE/CVF Conference on Computer Vision and Pattern Recognition (CVPR)*, 2019.
> [4] X. Xie, et al., "SNAS: Stochastic Neural Architecture Search," in *International Conference on Learning Representations (ICLR)*, 2019.
> [5] A. Ramesh, et al., "Zero-Shot Text-to-Image Generation," in *International Conference on Machine Learning (ICML)*, 2021.
> [6] M. J. Kusner and J. M. Hernández-Lobato, "GANs for Sequences of Discrete Elements with the Gumbel-Softmax Distribution," *arXiv preprint arXiv:1611.04051*, 2016.
> [7] W. Łańcucki, J. Majewski, P. Choromański, et al., "Robust Training of Vector Quantized Bottleneck Models," *arXiv preprint arXiv:2005.08520*, 2020.

---

> ### Comment · Reviewer_xGro · 2025-11-23
>
> Thanks, I have increased the contribution score to 3

---

### Official Review · Reviewer_7aj7 · 2025-10-30

**Soundness:** 2
**Presentation:** 4
**Contribution:** 3
**Rating:** 2
**Confidence:** 3

**Summary:**

In this work the authors advance a new program induction framework that involves the synthesis by a transformer of a series of tokens representing a program. The sequence is then put through a Gumble Softmax to enforce its discrete nature and then passed to the decoder, which is a recurrent network that processes each token in the program and evolves a particular state then produces an output. In order to ensure generalization, test time optimization is performed in order to optimize the program to match the given examples. This technique is significantly better than other similar techniques such as TTT or LPN when it comes to generalizing to programs of novel lengths or composing programs together in simple domains, and remains competitive in more realistic settings.

**Strengths:**

The idea of this paper is simple and clear

The results in Section 5.1 are compelling, and demonstrate that in contexts where generalization is paramount, NLI performs very well, and generates programs that are interpretable.

**Weaknesses:**

The main weakness of this paper is the fact that NLI’s outperformance of baselines is not demonstrated in any nontrivial domain. Section 5.4 demonstrates that NLI has competitive performance in real settings, but in no setting does it's unique generalization ability as described in Section 5.1 lead to an overall improvement. Ideally, some nontrivial domain or a modification of an existing one should be identified where compositionality and generalization are key so NLI can demonstrate its better performance. (e.g., filter an existing domain’s dataset such that the training dataset contains items of length <k and the test dataset contains items of length >k)

Minor:

334-335: use a term other than baseline performance here, “baseline" is also what you are using to describe the techniques you compare against.

**Questions:**

Why average across the different input embeddings? This seems to reduce the expressivity of the model.

---

> ### Author Response · Authors · 2025-11-21
> **Part 1**
>
> We thank the reviewer for their review and for acknowledging the strong generalisation results in Section 5.1 and noting the interpretability of programs. Below, we address both the discussed limitations and the question related to the use of mean embeddings. We have also updated the paper in response to minor edit suggestions.
>
> We first acknowledge that, after the strong generalisation capabilities shown in Section 5.1, performance is mixed across tests for different types of generalisation when scaling to the more difficult DeepCoder dataset. While the domains tested in section 5.1 are indeed simple, they are non-trivial, as they clearly demonstrate different generalisation capabilities across baselines, without confounding factors.. While scaling neural networks has led to impressive performance, the inability to perform simple compositional generalisation persists, as this benchmark shows. Secondly, we would like to highlight that the results on length generalisation, where NLI performs best, align with those from section 5.1, which specifically tested for this type of generalisation, and show NLI to be very strong at composing primitives into longer programs than those seen during training.
>
> However, we acknowledge that there are still limitations in the expressivity of the programs that NLI can represent. We have expanded our section on the limitations of NLI (Section 7) to clarify these further. This includes constraints such as single-token conditioning per layer and the sequential flow of execution. Without making further architectural advancements to address these capacity constraints, while maintaining compositional generalisation, NLI’s performance on real-world datasets will be constrained. However, we believe that NLI still represents a strong initial step towards scalable architectures capable of compositional generalisation. It introduces a new form of inference, leveraging Gumbel Softmax to approximate discrete sampling while executing the discovered discrete program representations token by token. This is a significant novelty compared to current architectures and is shown in section 5.1 to be capable of generalisation that current architectures are not, despite the simplicity of the benchmarks. We believe that future work should focus on increasing the expressivity of NLI while maintaining compositional generalisation performance, thereby moving NLI closer to applicability to larger real-world datasets.
>
> We do perform an additional experiment testing the impact of leveraging program representations during training for both NLI and LPN. The current base versions of these models demonstrate that they can achieve competitive performance only using input-output pairs during training. However, we also acknowledge that it limits the decoder's ability to learn useful primitives during training, as the encoder's induction capacity bottlenecks the decoder. We have therefore expanded our DeepCoder results to include variants of LPN and NLI trained *with* access to ground-truth programs during training (matching the assumptions of the other baselines, such as Transformer Program Synthesis, Latent Programmer, and ExeDec, while still using only 10% of the data on which those methods were trained). We have updated Figure 4 to reflect this, we highlight that NLI w/programs is only outperformed by Exedec across the test datasets, which makes use of a handcrafted domain specific language.
>
> We emphasise that the data seen at test time is still the same input-output pairs. While not a huge step change, this leads to a performance increase across the different test sets for NLI.
>
>
>
>
>
> | Subset                        | NLI   | NLI w/ Programs | LPN   | LPN w/ Programs |
> |-------------------------------|-------|-----------------|-------|-----------------|
> | Length Generalization         | 17%   | 20%             | 10%   | 5%              |
> | Compose Different Concepts    | 18%   | 21%             | 15%   | 16%             |
> | Switch Concept Order          | 13%   | 19%             | 16%   | 18%             |
> | Compose New Operation         | 22%   | 23%             | 22%   | 24%             |
> | Add Operation Functionality   | 14%   | 19%             | 23%   | 21%             |
>
>
> We provide a complete outline of how NLI training with two encoders was performed in Appendix D, each leveraging different data (programs and input-output pairs) simultaneously.

---

> ### Author Response · Authors · 2025-11-21
> **Part 2**
>
> **Response to Question regarding the use of Mean Embedding**
>
> Regarding the reviewer’s question on averaging over input embeddings, we choose this as it provides permutation-invariant pooling, which enhances robustness to input order variations. In addition, it enables the model to generalise to arbitrary specification sizes, as demonstrated in [1], something which in-context approaches struggle with. We acknowledge that more expressive aggregation schemes could be used and could lead to a stronger training signal for latent representations and the decoder, such as attention-based pooling, but note that these may overfit to the specification size and so leave this analysis for future work.
>
> [1] Macfarlane, Matthew V., and Clément Bonnet. "Searching latent program spaces." arXiv preprint arXiv:2411.08706 (2024).

---

> > ### Comment · Reviewer_7aj7 · 2025-11-22
> >
> > The change to the limitations section has reframed my view of the paper, and as such, the only weakness I identified no longer meaningfully applies. I have adjusted the score to a 6 accordingly.
> >
> > Length generalization is a reasonable justification for the mean aggregation operation, please include this in the text when introducing it.

---

> > > ### Author Response · Authors · 2025-11-30
> > >
> > > Thank you for acknowledging that our rebuttal and the updated limitations section successfully addressed your only concern. As suggested, we have added an explicit justification for the mean aggregation operation when it is introduced in Section 4.1, highlighting benefits of permutation invariance and enabling length generalisation with respect to the specification. This change is included in the revised manuscript. We thank the reviewer again for their valuable feedback.

---

### Official Review · Reviewer_Lj8j · 2025-11-03

**Soundness:** 3
**Presentation:** 3
**Contribution:** 3
**Rating:** 6
**Confidence:** 3

**Summary:**

The paper considers the program-by-example program synthesis setting. It presents the Neural Language Interpreter (NLI) which is an approach that aims to combine the complimentary strengths of symbolic and neural program synthesis methods. The aim is for NLI to learn its own discrete programming language from the training data.

NLI uses an Encoder-Decoder architecture, where the latent space represents a program and consists of T vectors (representing the maximum program length), where each vectors can be one of K values (representing the number of different symbols in the learned programming language). One of the K values is the “skip” command which allows for the generated program to be of varying length.
The encoder maps example input-output pairs to a program, while the decoder sequentially executes the generated program’s commands to make a prediction on a new input.

NLI makes use of the Gumbel-Softmax trick and is trained with a VAE-like objective, where the latent distribution regularization loss (referred to as the encoder loss) is designed to encourage primitives to be reused.

At test time, the encoder can be used to provide an initial prediction of the target program. Afterwards, due to the differentiability of the decoder, NLI can optimize its program using gradient optimization in order to improve its prediction accuracy.

The method demonstrates good program length extrapolation and good performance on novel composition tasks.

**Strengths:**

Getting around the limitations of symbolic methods, by learning a DSL appears to be a promising direction. The paper represents an important step towards this goal. The method appears to be novel and a creative combination of existing ideas. Apparently novel ideas include: the VAE-like architecture for program synthesis, the discrete program latent space, as well as the test-time gradient optimisation of the initially proposed program.

**Weaknesses:**

Presentation: There are two important details missing from the main text. First, the encoder loss is not specified in section 4, and only a high-level intuition is provided. This makes it difficult to evaluate whether this is a key part of the method and an important innovation. Second, when discussing the custom suite of benchmarks in section 5, an explanation of the basic operations is missing, making it difficult to evaluate the difficulty of the benchmarks.

While the limitations are discussed, a more comprehensive analysis would be beneficial to the paper. Currently, the paper comes across as a proof-of-concept that this is a promising direction, but the lack of clearly outlined limitations, coupled with limited evaluation weakens its broad claims.

**Questions:**

Could ground truth program representations be incorporated into the latent space supervision? Perhaps at a later stage of training?

Reconstruction loss: would it not help to also reconstruct all training pairs to make sure the program covers all examples?

---

> ### Author Response · Authors · 2025-11-21
> **Part 1**
>
> We thank the reviewer for acknowledging this work as an important step toward improving symbolic methods and for recognising the layers of novelty in this work. We aim to address all of the reviewers' questions and notes below.
>
>
> **Q1. Could ground truth program representations be incorporated into the latent space supervision? Perhaps at a later stage of training?**
>
> We thank the reviewer for this suggestion, which we agree is very promising. While our work aims to develop a method that does not rely on ground-truth program representations, such representations can be available in practice and could provide stronger supervision during training, beyond the partial signal from input-output examples alone. This could yield a stronger signal for learning the latent program representations and for training the decoder to execute them effectively. For instance, when only three input-output pairs are used, there may be high uncertainty over the encoded program, leading to weak or noisy signals for the decoder. In contrast, using the full program representation would provide a much richer signal to the encoder, ensuring the latent representation captures full information to pass on to the decoder for execution.
>
> One way to incorporate this is to use two encoders sharing a common latent space and decoder: one encoder for input-output pairs and another for tokenised program representations. This setup would enable two reconstruction losses to supervise the shared components. Only the encoder designed for input-output pairs would be used at test time to initialise the search.
>
> We have implemented the reviewer's suggestion into both NLI and LPN and tested performance on the DeepCoder datasets (e.g., using the tokenisation scheme from the DeepCoder paper, processed via a transformer). This brings our NLI w/Programs variant into line with training assumptions of the other baselines (Exedec,Latent Programmer and Transformer Program Synthesis) while continuing to train on 10x less data.. We stress that test-time assumptions and inference are identical for both NLI trained with ground-truth programs and the NLI baseline.
>
> | Subset                        | NLI   | NLI w/ Programs | LPN   | LPN w/ Programs |
> |-------------------------------|-------|-----------------|-------|-----------------|
> | Length Generalization         | 17%   | 20%             | 10%   | 5%              |
> | Compose Different Concepts    | 18%   | 21%             | 15%   | 16%             |
> | Switch Concept Order          | 13%   | 19%             | 16%   | 18%             |
> | Compose New Operation         | 22%   | 23%             | 22%   | 24%             |
> | Add Operation Functionality   | 14%   | 19%             | 23%   | 21%             |
>
>
>
> We see a boost in performance on NLI and LPN across the test sets when leveraging program representations during training. We add these results to the main body as we believe it to be a useful baseline using the same training assumptions as previous methods. In addition, we discuss the implementation of the Program Encoder and its training methodology in Appendix Section D.

---

> ### Author Response · Authors · 2025-11-21
> **Part 2**
>
> **Q2. Reconstruction loss: would it not help to also reconstruct all training pairs to make sure the program covers all examples?**
>
> Yes, it is correct that reconstructing all the training pairs is important to fully leverage available data. In Equation 1, the loss is averaged over all pairs for a given set of input-output pairs via a leave-one-out objective: n-1 pairs are encoded, and we decode the nth output, which is repeated across all examples for a given program. This is important, as the reviewer points out, to ensure the decoder can generalise across different inputs and get as much training signal as possible. The leave-one-out loss is essential to prevent trivial memorisation and encoding examples into the latent space to be decoded, preventing the learning of a well-structured latent space of programs.
>
> **Reviewers’ comments on adding encoder loss**
>
> The reviewer notes that the encoder loss is not specified in Section 4 but only in Appendix E, along with an explanation as to how this loss encourages sparsity of tokens. We have moved the encoder loss to the main body and have added an ablation to Figure 2 that removes this encoder loss, along with a discussion.
>
> **Reviewers’ comments on Benchmark information**
>
> We agree that the Deepcoder benchmark information is not explained in sufficient detail in Section 5 and have updated the main paper accordingly.
>
> **Reviewer's Comments on the Details of Limitations**
>
> We thank the reviewer for pointing out the need for a more comprehensive analysis of limitations, which would strengthen the presentation of our work, and have updated the limitations section of the paper to reflect this suggestion. While the current discussion highlights some key limitations, we acknowledge that expanding on these would better contextualise the method. To address this, we have expanded the discussion on limitations to include the absence of conditional branching, as well as potential training issues, e.g., vanishing or exploding gradients, from learning long programs. We also include a discussion in the experiment section and an ablation in the appendix on the importance of careful annealing in Gumbel-Softmax training; this is a critical yet underexplored aspect of our method that warrants explicit discussion in the paper.

---

### Official Review · Reviewer_MBC4 · 2025-11-04

**Soundness:** 4
**Presentation:** 3
**Contribution:** 4
**Rating:** 8
**Confidence:** 4

**Summary:**

This paper introduces Neural Language Interpreter (NLI), a neurosymbolic technique for program induction, which is based on having a neural network predict an intermediate program and interpreting it differentiably. Notably, the intermediate program representation is learned, so the architecture isn't fixed to a particular DSL! The predicted program is then refined using a neural executor. The paper then compares NLI to several baselines, establishing its general effectiveness. In particular, the paper emphasizes the ability for compositional generalization, and boasts strong results on OOD-generalization when compared to baselines.

**Strengths:**

The proposed method is original, sound, and well-described.

The OOD generalization results in Table 1 are compelling.

The ablation in section 5.2 solidifes the method, giving clear evidence for why it improves better than the baselines.

While NLI doesn't outperform every other method on the DeepCoder benchmark, performance is convincing.

Comparisons to existing work are done well, and the authors demonstrate a good understanding of the literature.

**Weaknesses:**

Further explanations of the learned intermediate language and differentiable interpreter would help the paper significantly (e.g. content from Appendix D could be partially reflected earlier in the paper, like in Section 5.1.1). I'd recommend partially using the extra page for camera-ready for this, since a lot of the paper's space is already well-allocated.

Commentary/explanations of DeepCoder benchmark performance would also help. While the results support some scalability, it seems further work might need to be done before the approach scales successfully, especially to more general programs (e.g. arbitrary Python programs).

The authors self-identified limitations, e.g. parameterized functions in particular (and especially limitations like this which effect expressivity).

**Questions:**

The paper makes an argument that the internal representations are more flexible and effective than hand-designed DSLs. Further analysis/experimentation of this point could prove interesting.

While an intermediate symbolic program is predicted, it isn't explicitly manipulated like the approaches cited in section 6 paragraph 1 would do. Accordingly, it seems that the overall approach still leans to neural processing. Have the authors considered doing symbolic manipulations of the intermediates? Since the intermediate representation is relaxed, this may not be explicitly possible, but it still seems like this representation could be interpreted/manipulated in some way. For example the analysis in Appendix D is quite interesting, and building on this could be a promising direction for future work.

---

> ### Author Response · Authors · 2025-11-21
> **Part 1**
>
> We thank the reviewer for their detailed review and for highlighting the originality of NLI, the strength of experiments in Section 5.1, 5.2 and comparison to baselines. We take the opportunity below to address questions and weaknesses.
>
> **Response to Q1:** “The paper argues that internal representations are more flexible and effective than hand-designed DSLs. Further analysis or experimentation on this point could indeed prove interesting.”
>
> We thank the reviewer for this insightful question and the opportunity to clarify this critical point.
>
> Carefully handcrafted DSLs can be highly effective within their targeted domains, often leveraging strong human-engineered priors to achieve excellent compositional generalisation (e.g., the FlashFill or DeepCoder DSLs). Thus, we do not claim that learned NLI representations will consistently outperform a well-designed handcrafted DSL.
>
> Nevertheless, handcrafted DSLs face two fundamental limitations that NLI directly addresses:
>
> 1. **Scalability**. Designing a high-quality DSL is a labour-intensive process that requires substantial human expertise and iterative refinement. In contrast, NLI is inherently scalable because the model learns its own discrete, symbolic-like programming language end-to-end from data, without any manual intervention. This self-supervised process allows rich representational structure to emerge automatically and to be optimised directly for downstream performance. Evidence of such emergent structure, comparable in systematicity to handcrafted DSLs yet discovered without human design, is provided in Appendix D. There, we show consistent reuse of learned primitives (e.g., tokens 231 and 476) across diverse out-of-distribution tasks.
>
> 2. **Test-time adaptability**. DSL-based approaches typically rely on domain-specific heuristic search (e.g., enumeration or genetic algorithms), which can be computationally expensive and brittle when applied to novel tasks. NLI, by contrast, learns a differentiable interpreter together with its language. This enables efficient gradient-based optimisation at test time, allowing programs to be refined directly on unseen specifications. The resulting gains in generalisation are demonstrated in Table 1, where NLI’s gradient-based search significantly outperforms heuristic-based baselines.
>
> In summary, while expert-designed DSLs remain powerful in narrowly scoped settings, NLI offers a more scalable and adaptive alternative that reduces human engineering effort and supports efficient test-time search.

---

> ### Author Response · Authors · 2025-11-21
> **Part 2**
>
> **Response to Q2:**
>
> We appreciate the reviewer's question, which gives us a chance to discuss the symbolic manipulation in NLI further.
>
> NLI inference can be viewed with the System 1 and System 2 framework. NLI System 1 is where the inductor directly predicts the program representation, and the interpreter executes it. We agree with the reviewer that this consists of a neural inference method and is not symbolic beyond the discrete bottleneck, contributing little to OOD generalisation on its own (e.g., 0% accuracy on Shift-L via direct prior sampling).
>
> In contrast, a core innovation in NLI is a System 2-style test-time search, which constitutes symbolic manipulation over the discrete program tokens. It refines the initial neural guess by optimising in the continuous relaxation of token sampling, exploring, and recombining learned primitives to fit the input specification. While the relaxation enables differentiability, similar to during training, we anneal the temperature to increase the discrete sampling approximation progressively.
>
> We did not pursue symbolic edits (e.g., rule-based token rewriting, as in cited works). We do, however, test discrete search with NLI called NLI prior search, which uses a pre-trained neural model to sample over the discrete program space. However, we found the gradient-based search to be very high-performing compared to such discrete sampling methods. That said, further work on possible search methods in the learned symbolic space is a promising direction for future work. In particular, finding a way to leverage gradients in such an approach could be a promising way to balance exploration and exploitation.
>
> **Response to further explanation on intermediate language representations and differentiable interpreter**
>
> We thank the reviewer for this constructive suggestion, which will undoubtedly improve the paper's accessibility. Integrating more details on the learned representations and interpreter earlier would benefit readers. We have moved the detailed discussion of program representations from Appendix D to Section 5.1.1.
>
> **Response to comment on scalability**
>
> The reviewer highlights that “further work might need to be done before the approach scales successfully, especially to more general programs (e.g. arbitrary Python programs).”
>
> We fully agree that further work is needed for broader scalability. As identified in Section 7, key limitations include support for parameterised functions and the need for larger token vocabularies to capture diverse computations. That said, these iterations on NLI are fully realisable and the topic of future research.
>
> We have improved the limitations section, expanding further on challenges for scaling NLI to more difficult problems. Despite these limitations, NLI establishes a strong foundation via its end-to-end learned discrete program representations and adaptation via differentiable search.

---

### Author Response · Authors · 2025-11-21
**Author Response Summary**

Dear Reviewers,

Thank you very much for your careful reading and highly constructive feedback. In addition to individual reviews, we provide below a summary review of the main changes to the manuscript in response to feedback.

All changes in the revised manuscript and appendix are highlighted in **blue**.

**Shared improvements across multiple reviews**

- Significantly expanded Section 7 (Limitations) with a more detailed and candid discussion
- Added explicit ablation to Appendix Section C and discussion in the main body on the importance of temperature annealing for Gumbel-Softmax training
- Clarified and better motivated the use of Gumbel-Softmax (vs. VQ-VAE straight-through, etc.) – now in the main paper in Section 5.3
- Moved several technical details from the appendix to the main body:
  - Learned program representations and interpreter (now Section 5.1.1)
  - Encoder regularisation loss + ablation
  - Detailed description of the DeepCoder benchmark and subsets

**New NLI variant added**

After feedback from reviewers, we trained an NLI variant that uses ground-truth program representations as additional supervision **during training only** (test-time protocol remains unchanged and still uses only 10% of the data of prior baselines). This is implemented via a separate program encoder, which we fully outline in Appendix Section D. Note that the usage of program representations to aid training brings “NLI w/ Program” in line with the training assumptions of ExeDec, Transformer Program Synthesis and Latent Programmer. We still include NLI without program representation results, as it is essential to highlight that NLI does not require these to learn.

This “NLI w/ Programs” variant shows small gains across all DeepCoder generalisation subsets, and LPN w/ Programs improves on 3 out of 5 subsets. NLI w/ programs is the strongest performing algorithm that does not make use of a handcrafted Domain Specific Language, second to Exedec which does use a DSL.

Best regards,
The Authors

---

### Author Response · Authors · 2025-12-03
**Rebuttal Summary**

Below we provide a brief summary of the rebuttal period for this paper.

Review 7aj7 had some initial concerns around the limitations outlined in the paper. During the rebuttal period we made significant changes to this section in the manuscript, resulting in Reviewer 7aj7 confirming that this additional information and clarity had positively reframed their view of the paper, acknowledging that this had addressed their only identified weakness.

Secondly, reviewer xGro had questions around the use of Gumbel Softmax in the paper. We provided a detailed summary of this design choice and its extensive use in the literature, which we included in the updated version of the manuscript. The reviewer acknowledged our response and had no further concerns following this.

---

### Meta-Review · Area_Chair_x1N2 · 2026-01-08

**Summary:**

1. Some algorithm details are not stated clearly.
2. Better explanation of the neural interpreter and the learned language.
3. Why using Gumbel-Softmax over VQVAE
4. No real-domain performance improvement.

**Reviewer Concerns:**

The authors have addressed the concerns of the reviewers, in particular the focus and scope of the work. Reviewers appreciate the idea of latent space representation for program synthesis, the quality of rebuttal and change scores accordingly.

**Reviewer Scores:**

MBC4: 8->8 (remain positive)
Lj8j: 6->6 (remain positive)
7aj7: 2->6 (misunderstand the contribution of the paper, and was convinced by the rebuttal)
xGro: 6->6 (remain positive)

---

### Decision · Program_Chairs · 2026-01-26

Accept (Poster)